# Protocol of the Budapest sleep, experiences, and traits study: An accessible resource for understanding associations between daily experiences, individual differences, and objectively measured sleep

**Wael Taji** (ID)**, Róbert Pierson, Péter Przemyslaw Ujma** (ID)*

Institute of Behavioural Sciences, Semmelweis University, Budapest, Hungary

* peteru88@gmail.com

## Abstract

Sleep is both a neurophysiological state and a biologically necessary behavior that performs a variety of indispensable roles for human health, development, and cognitive functioning. Despite this, comparatively little is known about the relationships between daily experiences and sleep features. Importantly, these relationships are bidirectional in nature, may be differently associated with subjectively and objectively assessed sleep, and may also be modulated by individual differences To address this challenge, we created the Budapest Sleep, Experiences, and Traits Study (BSETS), a multidisciplinary observational sleep study utilizing novel remote EEG devices. BSETS was designed to establish a dataset for future use in investigating the relationships between sleep features and daily experiences. In this paper we describe the protocol of the currently ongoing BSETS, which examines a community-dwelling sample of over 250 healthy participants who are studied in a naturalistic setting using a large questionnaire assessing psychological, demographic, and anthropometric information, as well as evening/morning diaries of sleep and daily experiences, and mobile EEG recordings over a period of 7 days. This dataset will become an accessible resource to the wider scientific community and can be utilized to investigate the complex multidirectional relationships between objectively and subjectively measured sleep, daily experiences, and individual differences, bestowing it with significant value for sleep researchers as well as practitioners working in clinical settings with patients suffering from disordered sleep.

## Introduction

Sleep in humans is a neurophysiological state with indispensable biological functions [1]. Sleep plays a role in human development [2, 3], somatic maintenance [4–6], as well as emotional regulation, cognitive problem-solving, and memory [7–9]. Interest in the importance of sleep and the challenges posed by deficient sleep and sleep disorders has grown significantly in

**Data Availability Statement:** Supplementary data and code used for the analyses described in this paper are available at https://zenodo.org/record/7799130. The Supplementary data also includes a

preliminary version of the BSETS database, complete with hypnogram data and bandwise EEG power. Information about daily events, potentially containing identifying information, was deleted, and variable names and responses in Hungarian were only translated for key variables. Additional information about data and (upon its completion) the full dataset will be made available to interested researchers. We ask those interested in publishing using BSETS data to consult with the authors of this paper before using the data.

**Funding:** PPU was supported by grant OTKA PD 138935, received from the National Research, Development and Innovation Fund of the Ministry of Innovation and Technology (https://nkfih.gov.hu/). The funders did not and will not have a role in study design, data collection and analysis, decision to publish, or preparation of the manuscript.

**Competing interests:** The authors have declared that no competing interests exist.

recognition over the last two decades [10]. The rise in general interest toward sleep can be seen in academic research settings [11] clinical settings [12], as well as in non-expert contexts.

How daily activities and experiences (DAEs) influence sleep, and are affected by sleep themselves has been a focus of interest since even before the birth of modern sleep research, but comparatively few studies have been published at least until the nineties [13]. There has been growing interest in this field recently, however, it is our view that methodological limitations persist in this field.

Studies frequently employ experimental study designs to establish the effects of DAEs on sleep. In these studies, a group of volunteers participate in some intervention (e.g. exercise) with a potential effect on sleep, and their sleep after this intervention is compared to a control condition. Recently, several reviews of studies with similar designs have been published. For example, one review [14] found some evidence for certain dietary patterns affecting sleep quality, but describe findings as preliminary. A recent meta-analysis [15] concluded that caffeine consumption substantially impacts multiple measures of sleep quality. Multiple reviews and meta-analyses [16–18] investigated the role of exercise on sleep, with somewhat divergent results. Experimental study designs are capable of establishing causality; however, they suffer from issues relating to breadth, statistical power and ecological validity. First, in experimental study designs only one DAE (for example, the consumption of a specific diet or a specific daily experience experimental participants are exposed to) can be tested per experiment, limiting breadth. Second, while there is no theoretical limitation to how many participants can be investigated in this design, because of the required rigor of the procedure the studies often end up being small, limiting power. Third, experimental manipulations may induce artificial conditions which imperfectly imitate naturalistic variation in DAE. For example, the experiment may consist of an extreme experience which is rarely or never seen in real life, which limits ecological validity.

Another study design which can elucidate the relationship between DAE and sleep are daily observational study designs, also known as naturalistic prospective studies. In this study design, sleep measurement and the measurement of DAE takes place each day and night for a period of time, typically at least one week. Using participants as their own controls, this study design is able to establish if sleep in the same person is systematically different after a DAE was experienced than otherwise. By studying differences in DAE as a function of the characteristics of sleep on preceding night, this design can also establish the effects of sleep on DAE. In a review of daily observational studies. [19] it was concluded that a bidirectional relationship between sleep and daily mood exists. Other studies revealed a generally bidirectional relationship between sleep and daily stress [20–22], and, recently, reduced sleep latency and possibly improved sleep quality after sexual activity [23]. Daily observational studies eliminate many of the shortcomings of experimental studies:

1. repeated measurements of the same person on multiple days establishes the causality (as causality can only flow from a night's sleep to the next day's DAE or from a day's DAEs to the next night's sleep and not vice versa)

2. the assessment of multiple DAEs is possible each day (for example, by an extensive questionnaire), eliminating the problem with breadth in experimental studies

3. as participants can follow their normal daily routines and do not have to adhere to a strict experimental regimen, problems with recruitment and protocol adherence are reduced and a larger sample with more statistical power can be achieved compared to experimental studies

4. for the same reason, daily observational studies are more ecologically valid than experimental studies.

The main limitation of current daily observational studies is the reliance on self-reports in sleep timing and quality assessments. While a handful of recent daily observational studies [22, 24] employed mobile EEG devices to establish the link between sleep and DAEs, this is not the case with most studies. While self-reports may provide valuable information about sleep, the accuracy of self-reports are at best moderate [25], and the estimation of sleep macrostructure (such as the proportion of NREM and REM) is not possible based on self-reports.

A possible general limitation of both experimental and daily observational studies is ignoring the role of between-individual variation. Sleep features across multiple nights tend to be highly similar within the same individual [20, 22, 26–28], in a large part due to genetic reasons [29–32]. The same is true for daily experiences ([20, 22, 24]), which are not random but in part follow the trait-like predispositions of individuals and tend to be themselves highly heritable [33–36]. Therefore, studies of the relationship between daily experiences and sleep routinely analyze within-individual (implying causality) and between-individual (implying a common cause or correlated reporting bias) effects separately [19].

We established the Budapest Sleep, Experiences, and Traits Study (BSETS) to study the bidirectional relationship between DAEs and sleep. BSETS opts for a daily observational study design where participants are studied observationally (without intervention by researchers that could affect or bias participant behavior and/or results) over a 7-day and night period. A minimally intrusive observational study design is a key feature of BSETS, entailing a data collection period of 7 consecutive days and nights during which both objective and subjective measures of sleep, as well as an extensive list of subjective measures of daily activities and experiences, are recorded. To overcome the limitation of subjective sleep measurements, BSETS utilizes both objective (using a mobile EEG headband) and subjective (using morning diaries) measurements of sleep. To address the possible role of between-individual variation, we included a detailed battery consisting of psychometric and anthropometric measures in BSETS to provide a broad measurement of trait-like characteristics. We this, we first aimed to investigate what other traits sleep features, themselves highly trait-like, can be associated with. For example, typical sleep features of individuals (measured as the average of the seven nights) may be associated with personality traits [37], intelligence [9], or subclinical symptoms of psychiatric illness [38, 39]. Second, we aim to use trait measures as moderators in analyses of within-individual associations. For example, it is possible that trait neuroticism moderates the degree to which stressful events during the day impact sleep on the subsequent night, or that the effect of sexual intercourse on sleep quality is different in men and women [23].

In the current paper, we describe the protocol of BSETS. While some empirical findings are presented in this paper, the main quantitative analyses of BSETS will follow upon completion of the dataset. By using questionnaire-based trait and ability measures, self-reported daily and nightly experiences, as well as objectively measured (EEG-based) sleep measures, in the future we will be able to unravel:

1. Sleep EEG biomarkers of self-reported psychological characteristics and abilities.

2. The effect of self-reported daily mood and experiences on objective and subjective sleep characteristics during the following night.

3. The effect of subjective and objective sleep characteristics on self-reported mood and experiences during the following day.

## Materials and methods

To achieve our primary research goal - that is, to create a database that can be used to explore relationships between traits, objectively and subjectively assessed sleep measures, and daily experiences, a study design and methodology was carefully selected with a view of circumventing the common limitations detailed above. A minimally intrusive 7-day observational study protocol using two daily self-reports of daily/nightly experiences, and portable worn EEG devices were utilized to facilitate the study's emphasis on remote measurement in a naturalistic sleeping environment of participants' home. The study design, measure selection, and data management protocols are described below.

### Study design, setting, and participant recruitment

The BSETS is an observational study in an ecologically valid setting that monitors participants for 7 consecutive days and nights, with preliminary assessments taking place online as well as in the laboratory where devices are handed over. All in-person assessments take place at a single center in the NET Building of Semmelweis University, in Budapest, Hungary.

The study is designed to be a cross-sectional, observational, quantitative study of volunteers, with an approximate total sample size of N = 250 participants, who are recruited via the method of branched diffusive convenience sampling. The selection criteria are: healthy volunteers over the age of 18 who are native speakers of the Hungarian language to ensure the comprehension of research protocols and requirements and capable of providing informed consent. Recruitment from clinics or clinical settings was deliberately avoided due to concerns about the generalizability of the sampling pool, as the primary purpose of the BSETS dataset is the investigation of sleep-DAE-trait relationships as opposed to sleep disorders or the effect upon these relationships of medical conditions that are implicated in disordered sleep. We collect data about daily medication use, permitting the post-hoc exclusion of participants seriously affected enough to be medicated. Although BSETS is an ongoing research study, we present summary statistics describing the features of our participant pool (e.g. sex ratios, average age, average level of educational attainment, test results) in the results section; further summary statistics will be made available as the study progresses.

Participant recruitment was initialized through the use of branched diffusive convenience sampling, for which candidates for research participation were solicited through in-person interactions, via email, and through solicitations publicized in specific online forums. No financial incentives or rewards were offered in return for research participation; compensatory incentives were restricted to the release of processed sleep data in the form of an automated weekly sleep report to participants following the completion of the study, as well as student credits in the case of participants or the students recruiting them already enlisted in pre-listed courses at Semmelweis University. Participants were encouraged informally to invite their family members, friends, and romantic partners to participate in the study following their own successful completion in our research. However, multiple members of the same household never participated in the study at the same time.

Data collection started in October 2021, and is expected to conclude in May 2023.

### Ethics

The Institutional Review Board (IRB) of Semmelweis University as well as the Hungarian Medical Council (under 7040-7/2021/ EÜIG "Vonások és napi események hatása az alvási EEG-re" (The effect of traits and daily activities and experiences on the sleep EEG)) approved BSETS as compliant with the latest revision of the Declaration of Helsinki [40]. All participants gave written informed consent on a form reviewed and approved by the IRB.

During data collection, authors had access to information which could identify participants. This was necessary because personal contact via email was required to deliver and collect questionnaires and login data for the mobile EEG application. However, data is de-identified by using an internal code system after data collection. We only publish de-identified data from BSETS.

## Measures, questionnaires, and outcomes studied

BSETS utilizes four discrete measurement domains for collecting data on sleep, traits, and experiences from participants prior to and during the 7 days and nights of the study period.

1. Psychological Trait Measures (or Traits); gathered via pre-study psychological testing

2. Objective Sleep Measures; gathered via portable EEG devices

3. Subjective Sleep Measures; gathered via participant self-report

4. Daily Activities and Experiences (DAEs or Experiences); gathered via participant self-reports

Detailed descriptions of the measures, tests, and tools utilized are provided below.

**Psychological trait measures.** In BSETS, the use of extensive pre-trial psychological test batteries is a means by which to establish a holistic snapshot of participant psychophysiological profiles. The role of these is both to serve as potential correlates of sleep themselves, but also to better contextualize findings related to day-to-day variation as moderators. When evaluating the suitability and significance of pre-trial psychological tests for inclusion in BSETS, the following criteria were applied:

1. The test must measure a trait with an established relationship with components or features of sleep, and a history of use in sleep research.

2. The test must be validated by previous studies.

3. The test must be intelligible to non-specialist general audiences and must be completable within reasonable time without researcher intervention or assistance.

Upon applying these criteria, we selected 10 psychophysiological test batteries and a further custom cognitive test that were selected for inclusion. These tests, the context of their use, their measurement target, their relevance for BSETS, and their associativity with sleep are described in Table 1 below.

**Daily Activities and Experiences (DAEs): Evening diaries.** The evening diaries describe the DAEs of participants prior to sleep and at the end of a finished day. They are required to be filled out each night before sleep. Similar to the morning diaries, the structure of the evening diaries can be broken down into two sections.

*Day Descriptions*: here participants provide basic information about their day, and evaluate it in terms of emotion and affect (time required: ≤ 3 minutes) (**Table 2**).

*Daily Experiences and Events*: here participants respond to a series of common events and situations, and describe whether or not they experienced these in the preceding day (time required: ≤ 7 minutes) (**Table 3**).

7 identical copies of evening diaries are provided to each participant. Because university students make up the majority of the participant population, BSETS researchers anticipated that poor compliance in terms of diary completion timing might be higher due to the social schedules and nightlife habits common among student populations. The evening diaries

**Table 1.  A summary of trait test batteries used in BSETS.**

| Test battery and abbreviation | Test Administration Setting and Context Use case/rationale Test Use and Description | Measurement target in the context of BSETS study goals | Rationale for inclusion | References |
|---|---|---|---|---|
| **44-item Big Five Inventory (BFI-44)** | Used in clinical and psychiatric settings to screen for personality disorders and psychopathological traits | **Personality** (Openness, Conscientiousness, Extraversion, Agreeableness, Neuroticism) | Personality is known to correlate with chronotype and preferences as well as DAEs; it is also extensively correlated internally with many of the other demographic variables of use in our study | [41] **Hungarian version:** [42] |
| **99-item Zuckerman-Kuhlmann Personality Questionnaire (ZKPQ)** | Is used both in isolation and in conjunction with BFI-44 for personality assessment. | **Personality** within a non-BF framework | ZKPQ is a personality assessment tool developed from theoretical principles in a top-down fashion, unlike the BFI which was developed using factor analysis from the bottom-up. Incorporating a secondary, theoretically-driven layer of personality assessment in the form of the ZKPQ into our test batteries allows for more robust analysis of associative findings linking sleep or experiences with personality | [43] **Hungarian version:** [44] |
| **20-item Toronto Alexithymia Scale (TAS-20)** | Diagnostic tool to assess the severity and degree of dissociative symptoms and the risk of associated socio-emotional dysfunction | **Dissociative symptoms** | Alexithymia is a recognized mental-psychological disability or debilitating personality characteristic not listed in the current version of the DSM (DSM-V). This condition or trait is known to have profound and dramatic consequences on individual behavior and by extension, as well as independently, on sleep features also. The TAS-20 was therefore considered important for our study as it allows for a rare but impactful constellation of psychological features to be spotted where other tests may detect this constellation only partially. | [45, 46] **Hungarian version:** [47] |
| **28-item Dissociative Experiences Scale (DES-II)** | Screening tool for psychotic disorders such as schizophrenia, as well as for trauma-induced dissociation syndromes | **Dissociative symptoms** | Dissociativity is a normal psychological trait with variation in the normal population, but clinically significant symptoms of dissociativity appear frequently in psychotic disorders such as schizophrenia, and are known to influence sleep behaviors and sleep features on the EEG, as well as daily experiences and activities. The DES-II was therefore included to allow for the comparison of these outcomes with dissociative symptoms. | [48] **Hungarian version** [49] |
| **9-Item Patient Health Questionnaire (PHQ-9)** | Clinical tool used to objectify the severity and symptomatology of depression most common 9 symptoms of depression and depressive disorders | **Depressive symptoms** | Depression and depressive disorders have known and significant correlative relationships with sleep and. The PHQ-9 was included so as to allow for these relationships to be studied within BSETS, and to search for further multilateral associations between sleep, experiences, and depressive symptoms | [50] |
| **6-item Edinburgh Handedness Inventory (EHI)** | Used in a variety of clinical and non-clinical contexts to assess left-right handedness, hemispheric laterality, and hand-hemisphere disconnectedness | **Left-right handedness**, and by extension **hemispheric laterality** | Left-right handedness is a known indicator of hemispheric lateral dominance, which itself is a significant indicator of sleep features in EEG recordings. The EHI was included to allow for participant features in sleep EEG recordings to be better contextualized and understood. | [51] |

(*Continued*)

**Table 1.** (Continued)

| Test battery and abbreviation | Test Administration Setting and Context Use case/rationale Test Use and Description | Measurement target in the context of BSETS study goals | Rationale for inclusion | References |
|---|---|---|---|---|
| **Munich Chronotype Questionnaire (MCTQ)** | Used to determine chronotype and chronotypical abnormalities | **Chronotype (individual biological clock)** | Chronotype is self-evidently a major determinant of sleep features (most obviously sleep timing) and sleep preferences. The MCTQ was included to allow for individual chronotype to be evaluated in relation to sleep features of participants.. | [52] **Hungarian version:** [53] |
| **8-item Athens Insomnia Scale (AIS)** | Clinical and diagnostic tool used to evaluate presence of insomnia symptoms, and severity of symptoms to patients | **Insomnia symptoms** and **insomnia symptom severity** | Insomnia is among the most ubiquitous features of disordered sleep, in both pathological and healthy populations; its measurement can be considered indispensable for any large-scale sleep study of diverse non-clinical populations. The AIS was included to quantify the influence of insomnia in sleep features, and to better understand how these features may be ascribable to either traits, DAEs, or both. | [54] **Hungarian version:** [55] |
| **16-item International Cognitive Ability Resources Intelligence test (ICAR-16)** | An open-source tool used internationally and largely outside of clinical contexts to assess cognitive abilities (IQ) | **Cognitive ability (IQ)** | Intelligence (operationalized as IQ) has known correlates with sleep features. Some of these, most notably the correlation between IQ and sleep spindle activity during NREM sleep, have been extensively described by EEG researchers, while others may remain unknown or undiscovered. IQ likely also plays a role in DAEs, though this is presently unquantified. The ICAR-16 was included so that the influence of intelligence on sleep features could be more closely scrutinized, and compared with other individual traits as well as DAEs. | [56–58] |
| **"People Test" online adaptive reasoning, vocabulary and spatial ability test series** | An adaptive test with a logical reasoning (24 items), spatial reasoning (12 items), and vocabulary (24 items) component. | **Cognitive ability (IQ)** | There are established links between intelligence and certain aspects of sleep. EEG studies have extensively documented the association between IQ and the presence of sleep spindles during NREM sleep. However, there may be other links between IQ and sleep that have not yet been identified. As ICAR-16, this also was included To conduct a more detailed examination of how intelligence affects sleep patterns, it would be useful to compare it with other personal characteristics, daily routines, and experiences. | [59] |
| **8-item custom letter and number series test** | On the grounds of its purpose, it is not used in practice currently. | **Cognitive ability (IQ)** | This test was added by researchers as a means by which to experimentally assess its validity against the other IQ instruments. As such, this test has no reference in the dataset. | |

therefore begin with a quality control question, asking participants if they filled out the questionnaire in the evening as required, or in the morning from memory.

After this quality control question, participants provide a free-form description of the day.

After this, the DAEs section of the evening diaries begins, starting with a prompt listed below (Table 3).

**Table 2. A summary of the evening diary sections used in BSETS for the overall description of the previous day.**

| # | Question | Answer format |
|---|----------|---------------|
| 1 | Did you fill out this diary in the evening as required, or are you filling it out the morning after waking up from memory? | 1. In the evening.<br>2. The next morning, from recollection |
| 2 | Assess how closely the following statements correspond to your daily experience, on a scale ranging from 0 (absolutely not like this) to 10 (very much like this). | 1. Was your day physically exhausting?<br>2. Mentally exhausting?<br>3. Interesting and eventful?<br>4. Happy? |
| | PANAS (The Positive and Negative Affect Scale) daily ratings | Participants are asked to rate and evaluate the day in terms of affect using the short form [60] of the Positive and Negative Affect Schedule, a Likert scale leveled from 1 to 5 [61]. |

**Objective sleep measures: EEG recordings, the Dreem 2 EEG headband, and monitoring.** In BSETS, EEG constitutes the primary method for measuring sleep quality and sleep features over a full night of sleep. In order to obtain sleep EEG recordings from research participants under naturalistic conditions with minimal invasiveness, we made a decision to utilize a novel remote EEG device, the Dreem 2 EEG headband (**Fig 1**), that can be used at home with a minimal level of researcher instruction.

The Dreem 2 EEG Headband device, as described in the corporate whitepaper [62] measures EEG signals with 3 electrodes in the frontal band (prefrontal position) and two more at

**Table 3. A summary of the evening diary sections used in BSETS for the description of specific experiences during the previous day.**

| # | Question | Answer Format |
|---|----------|---------------|
| **Prompt:** Which of the following did you experience during your day? For "I experienced this" please answer 1, for "I did not experience this" please answer 0. | | |
| 1 | I did sports or an athletic activity for at least 30 minutes. | Yes/no. |
| 1a | If so, what kind of athletic activity did I participate in? | Freeform writing. |
| 2 | I slept or had a nap during the day. | Yes/no. |
| 2a | If so, how many times, and how minutes in total? | Freeform writing. |
| 3 | I spent at least 30 minutes in the company of people other than the ones I live with. | Yes/no. |
| 4 | I was in school or at the university. | Yes/no. |
| 5 | I was at a workplace. | Yes/no. |
| 6 | I consumed alcohol. | Yes/no. |
| 7 | I watched movies or tv series for over an hour (passive watching) | Yes/no. |
| 8 | I spent at least an hour in front of a computer. | Yes/no. |
| 9 | I was reading or playing on a cellphone or another smart device. | Yes/no. |
| 9a | If the answer is yes, then for how many minutes. | Freeform writing (total time). |
| 10 | I had sexual contact with someone. | Yes/no. |
| 11 | I drove a car or another vehicle for at least 1 hour. | Yes/no. |
| 12 | I spent at least 1 hour traveling by public transportation. | Yes/no. |
| 13 | I had a serious conflict or fight with someone. | Yes/no. |
| 14 | How much time did I spend outside, including transportation, but only if it was on foot (walking around outside). | Freeform writing (total time). |
| 15 | Did I take any medication, including normal habitual medication? | Yes/no. |
| 15a | If so, what medications did I take? | Freeform writing (medications). |

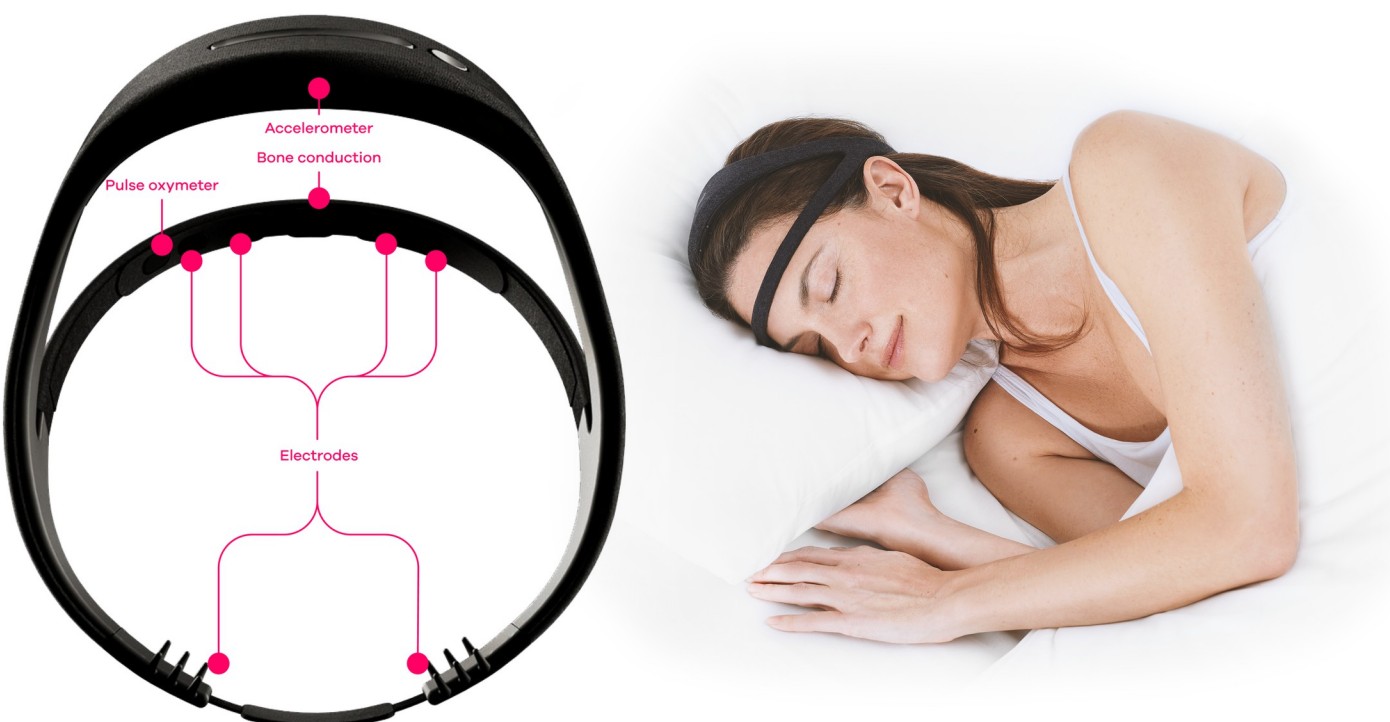

**Fig 1.** Ilustration of the Dreem 2 EEG Headband, with the layout of the device (left) and a person illustrating the way it is worn (right).

the back of the head in the occipital position, measuring channels F7-O1, F8-O2, Fp1-F8, F8-F7, and Fp1-F7. An additional EEG electrode is used as a bias. Signals can be acquired through the medium of user hair, and a mechanical system distributes the pressure of electrodes to minimize discomfort. The sampling rate of EEG recordings is set to 250 Hz. By default, recordings are band-pass filtered to 0.4–18 Hz. A pulse oximeter measures the blood oxygen saturation and heat rate via reflectance oximetry. The device is controlled via the Dreem Coach mobile application that is available for both Android and iOS smartphones.

Participants are carefully instructed in the correct use of the Dreem 2 EEG Headband devices prior to the conclusion of the handover process; this includes instruction in the correct method of wearing, activating, deactivating, storing, and charging the devices, as well as in the method of connecting them to Wi-Fi and to the mobile application (Dreem Coach) used to operate the Dreem 2 EEG Headband. Participants are instructed to use the Dreem Coach mobile application to turn the devices on prior to sleep, or to restart the device if turning it off and then on again manually during the night. During data acquisition, a research assistant is available 24/7 to participants to respond to questions and solve problems that emerge during work with the headbands.

EEG recordings obtained from wearable remote EEG devices (Dreem 2 EEG headband) are uploaded to a cloud-storage database after the conclusion of each night of sleep and are inspected manually by researchers each morning. If the quality of recordings is judged as suboptimal, participants are contacted again with a reminder about the proper way to use the Dreem 2 EEG headband.

EEG recordings are scored automatically using an algorithm which has demonstrated high consensus with manual scorings using standard criteria of the American Academy of Sleep Medicine [63]. This algorithm also calculates sleep macrostructure metrics, including total

sleep time, sleep onset latency, wake after sleep onset, sleep efficiency and the duration and percentage of wakefulness, N1, N2, N3, total NREM and REM in the recordings.

Artifactual epochs in EEG recordings are detected using a complimentary algorithm devised by Dreem Inc., which has been trained to reproduce the visual scorings of experts [62]. Each 2-second epoch, on each channel, is given a quality rating which corresponds to the likelihood of that epoch not being artifactual. Using these quality scores enables the exclusion of artifactual data from analyses on a channel wise basis. Several definitions could be used for identifying which epochs are to be excluded; of these, we have explored the effect of several on the tradeoff between data quality and the quantity of available data (see Results). In short, the determination was made to exclude data epochs if they did not reach a certainty of either 50 or 75% of not being artifactual, and all data from channels was also discarded if channel quality (the proportion of epochs from that channel scored as non-artifactual with at least 50% certainty) did not reach 20% or 50%. We excluded vigilance states from analyses if there were fewer than 20 epochs from that vigilance state fulfilling quality criteria from a channel. This aspect of the quality control process is necessary because even with high signal quality the number of data epochs fit for analyses can be scarce if a vigilance state (for example, REM) was rarely detected, resulting in noisy estimates of EEG metrics.

The availability of hypnograms and artifact scorings enables the use of standard time-series analysis methods, including the calculation of power spectral density (PSD), between-channel coherence and the detection of EEG oscillations. Of which, we elected to implement PSD analysis, and as a proof of concept. We attempted to replicate the well-known negative relationship between age and power in the delta and sigma frequency ranges, representing the loss of slow waves and sleep spindles in older individuals. PSD calculations were performed using EEGLAB's periodogram() function with 2-second nonoverlapping epochs and Hamming windows.

**Subjective sleep measures: Morning diaries and subjective self-reported assessments of sleep.** The morning diaries are required to be filled out each morning after waking up and can be completed in 5 to 10 minutes. The morning diaries can be subdivided into two sections (Table 4).

*Sleep Diary Section:* here participants provide subjective self-reports on the context, setting, and quality of sleep (time required: $\leq$ 5 minutes).

Table 4. BSETS morning diary, sleep diary section.

| # | Question | Answer format |
|---|----------|---------------|
| 1 | With whom (if anyone) did you sleep? | 1. I slept alone<br>2. I slept with my partner<br>3. I slept with a friend or another person<br>4. I slept with a pet |
| 2 | Where did you sleep? | 1. In my usual place (e.g. *a bed at home*)<br>2. At home, but not in my usual place (*e.g. on the sofa etc.*)<br>3. At the house of my friend or partner<br>4. In a hotel or another place where I paid to stay<br>5. Other: (*freeform writing entry*) |
| | Groningen Sleep Quality Scale subsection | Here participants complete the GSQS-15 [64] a 15-item battery of sleep complaints that provides a score of global sleep quality between 0–14, where 14 indicates exceptionally poor sleep quality. |
| 3 | When did you go to sleep yesterday evening? | Freeform writing entry (time of day). |
| 4 | How much time did it take for you to fall asleep, in minutes? | Freeform writing entry (approximate amount of time to fall asleep). |
| 5 | When did you wake up this morning? | Freeform writing entry (time of day). |

***Dream Diary Section:*** here participants who affirm the recollection of dream experiences occurring in the preceding night describe their dreams and their contents (time required ≤ 7 minutes).

7 identical copies of these diaries are provided to each participant, each containing an instructive preamble to remind participants when and how to complete the diaries (the main requirement being that they are filled out on all seven mornings of all seven days of the study, in the morning, not long after waking up). Participants record the time they begin to fill out the diaries, before answering a series of questions, which are as follows (Table 4).

After the Sleep Diary Section is completed, participants are asked to report their dream experiences using a recently developed dream affect inventory [65]. First, they are asked to respond to the following question (#6) with yes or no:

*"Please read the following statement, and answer with a 'true' or 'false' as best applies to yourself*:

*'I remember what kind of dream I had, what I was thinking about, and what feelings I felt before I woke up.'*

*In this question, by dream, we mean any experience, thought, or memory which crossed your mind when you were sleeping. This question applies to dreams in the morning, but also dreams you remembered when waking up during the night."*

Participants who respond "yes" to this question are prompted to continue with the dream component of the morning diaries (**Table 5**). Those who do not are requested to skip these questions.

This final question in the dream diary section, in which participants write as little or as much as they wish about their dreams; keywords, sentences, or complete paragraphs, enables semantic analysis and natural language processing tools to be applied to participants' descriptions of dream content. In doing so, we can gauge the emotional valence of dreams, and

**Table 5. BSETS morning diary, dream diary section.**

| # | Question | Answer format |
|---|----------|---------------|
| 7 | To what extent did the following feelings characterize your dream experience? | Happiness (1–5 Likert scale) |
| | This list of emotions is based on a recently published dream affect inventory [65]. | Guilt (1–5 Likert scale) |
| | | Contentment/satisfaction (1–5 Likert scale) |
| | | Surprise (1–5 Likert scale) |
| | | Fear (1–5 Likert scale) |
| | | Curiosity (1–5 Likert scale) |
| | | Sadness (1–5 Likert scale) |
| | | Security (1–5 Likert scale) |
| | | Anger (1–5 Likert scale) |
| | | Shame (1–5 Likert scale) |
| | | Insecurity (1–5 Likert scale) |
| 8 | (Standardized) In total, how do you emotionally rate your nightly dream experience? | Liker scale between 1 (very negative, scary, unpleasant, oppressive) and 9 (positive, happy, liberating). |
| 9 | With a few short sentences or keywords, please describe your dream or your dreams that you can recall after waking up today. | Freeform writing. |

conformity of dream content with participant's descriptions of sleep quality, of daily experiences, or with EEG-evaluated sleep features.

**Data and code availability.** Supplementary data and code used for the analyses described in this paper are available at https://zenodo.org/record/7799130. The Supplementary data also includes a preliminary version of the BSETS database, complete with hypnogram data and bandwise EEG power. Information about daily events, potentially containing identifying information, was deleted, and variable names and responses in Hungarian were only translated for key variables. Additional information about data and (upon its completion) the full dataset will be made available to interested researchers. We ask those interested in publishing using BSETS data to consult with the authors of this paper before using the data.

## Results and discussion

### Descriptive statistics

Data collection for BSETS is still ongoing. Here, we present preliminary results about data availability, sample demographics and quantitative EEG metrics.

So far, 208 volunteers took part in BSETS, of which 108 were females and 93 males. The gender of 7 participants remained unknown, as they did not respond to the trait questionnaires sent to them. The mean age in the sample was 28.83 years (SD = 12.49 years, range: 18–76 years). The age distribution of the sample was highly skewed, however, with 59.8% of the sample being younger than 25 years old, 11.3% being 26–35 years old, 6.7% 36–45 years old, 15.46% 46–55 years old and 3.1% older than 55 years old. The age distribution of the sample reflects the fact that most participants were friends or parents of medical students. 36.87% of the sample reported having advanced education (BA, MA or equivalent). 48.99% reported having a high school diploma, while 4.05% reported having vocational high school degrees. 5.56% of the sample reported only having completed eight grades of schooling, while 4.55% reported having a postgraduate degree. In a separate question, 56.6% reported being university students, likely representing students who will eventually obtain a university degree. Detailed descriptive statistics are provided in **Table 6**.

In total, we recorded 1439 nights from participants in BSETS. We have registered 1383 diary reports (96.1% compliance rate) and 1399 EEG recordings (97.2% completion rate) from these nights. Some participants are still expected to return diaries in the future, which would further increase the diary compliance rate.

200 participants provided at least one night of EEG data. Some EEG recordings were lost due to technical problems. These usually happened because the recordings didn't start or weren't started by participants, or data was lost during the process of cloud uploading due to device errors. These nights were treated as missing data.

The mean number of nights per participant exceeds 7, which is the normal duration of the research protocol. This is because chose to continue the study beyond the original seven days. In the dataset, we included diaries for these additional nights, increasing the number of nights recorded from these participants. Furthermore, because of multiple missing recordings, typically due to the failure of the EEG device participants were wearing, five participants were requested to take part in the study for a second week. Our day-to-day analyses assume that all data comes from subsequent days and nights. This assumption is violated if participants do not repeat the study on subsequent weeks. This was the case for four of these participants, from whom partial data from the first week was discarded and only the second week was used. The fifth participant, however, took part in the study on two consecutive weeks: therefore, we included data from all 14 nights (some with missing EEG data, but all with valid diaries) from this participant.

**Table 6. Descriptive statistics of key variables in BSETS.**

| Key Variables in BSETS | N | Minimum | Maximum | Mean | SD | Skewness | Kurtosis |
|---|---|---|---|---|---|---|---|
| Total sleep time | 199 | 136,00 | 525,14 | 393,63 | 58,49 | -0,78 | 1,91 |
| Sleep onset latency | 199 | 4,07 | 66,29 | 15,36 | 10,17 | 2,09 | 5,52 |
| Wake after sleep onset | 199 | 1,92 | 175,00 | 21,73 | 18,61 | 4,17 | 26,95 |
| Wake duration | 199 | 12,14 | 224,50 | 40,05 | 25,56 | 3,53 | 19,08 |
| N1 duration | 199 | 7,58 | 68,43 | 23,88 | 8,33 | 1,43 | 4,10 |
| N2 duration | 199 | 75,50 | 310,14 | 183,67 | 41,86 | 0,15 | 0,01 |
| N3 duration | 199 | 0,00 | 144,58 | 83,27 | 26,72 | -0,08 | -0,09 |
| REM duration | 199 | 42,50 | 204,50 | 102,82 | 23,95 | 0,05 | 1,09 |
| NREM duration | 199 | 93,50 | 388,79 | 290,81 | 46,51 | -0,49 | 1,30 |
| N1 percentage | 199 | 2,33 | 14,64 | 6,13 | 2,03 | 1,50 | 3,18 |
| N2 percentage | 199 | 28,76 | 62,65 | 46,03 | 6,78 | 0,11 | -0,25 |
| N3 percentage | 199 | 0,00 | 40,29 | 21,95 | 7,13 | -0,04 | -0,03 |
| REM percentage | 199 | 12,85 | 43,96 | 25,89 | 4,84 | 0,56 | 1,53 |
| NREM percentage | 199 | 56,04 | 87,15 | 74,11 | 4,84 | -0,56 | 1,53 |
| N2 latency | 199 | 0,20 | 17,58 | 4,34 | 2,57 | 1,83 | 5,30 |
| N3 latency | 198 | 6,00 | 92,25 | 18,49 | 9,98 | 3,06 | 16,01 |
| REM latency | 199 | 19,00 | 161,80 | 79,29 | 22,44 | 0,95 | 1,13 |
| Awakenings | 199 | 4,33 | 47,71 | 19,44 | 7,38 | 1,02 | 1,37 |
| Sleep efficiency | 199 | 37,66 | 96,62 | 90,64 | 6,02 | -4,58 | 33,09 |
| PANAS positive | 192 | 8,43 | 22,71 | 16,05 | 2,99 | -0,07 | -0,13 |
| PANAS negative | 192 | 5,00 | 14,57 | 7,93 | 2,12 | 0,83 | -0,01 |
| Sleep onset, workday (MCTQ) | 193 | -2,97 | 9,42 | 0,03 | 1,29 | 2,16 | 13,88 |
| Sleep onset, free day (MCTQ) | 194 | -2,58 | 5,53 | 0,58 | 1,28 | 0,61 | 1,86 |
| Getting out of bed, workday (MCTQ) | 192 | 5,00 | 14,17 | 7,26 | 1,27 | 1,48 | 4,80 |
| Getting out of bed, free day (MCTQ) | 192 | 0,75 | 16,00 | 9,16 | 1,79 | -0,07 | 3,56 |
| Sleep duration, workday (MCTQ) | 191 | 3,08 | 9,75 | 7,00 | 1,06 | -0,27 | 0,61 |
| Sleep duration, free day (MCTQ) | 193 | -2,75 | 11,17 | 8,13 | 1,38 | -2,51 | 18,87 |
| Total bedtime, workday (MCTQ) | 192 | 4,67 | 11,00 | 8,05 | 1,07 | 0,07 | 0,56 |
| Total bedtime, free day (MCTQ) | 192 | 0,75 | 13,00 | 9,49 | 1,42 | -1,05 | 6,42 |
| Midsleep, work day (MCTQ) | 191 | 1,02 | 11,71 | 3,54 | 1,15 | 2,22 | 13,07 |
| Midsleep, free day (MCTQ) | 193 | 1,33 | 10,52 | 4,65 | 1,38 | 0,55 | 2,38 |
| MCTQ_sdweek | 128 | 5,05 | 9,36 | 7,25 | 0,84 | -0,16 | -0,25 |
| Light exposure per week, MCTQ) | 113 | 10,00 | 360,00 | 56,44 | 44,75 | 3,40 | 18,63 |
| Relative social jetlag (MCTQ) | 191 | -9,50 | 5,31 | 1,13 | 1,18 | -3,70 | 34,42 |
| Absolute social jetlag (MCTQ) | 191 | 0,00 | 9,50 | 1,27 | 1,03 | 3,04 | 21,29 |
| Chronotype (MCTQ) | 140 | 1,33 | 8,06 | 4,31 | 1,26 | 0,16 | 0,32 |
| Weekly sleep loss (MCTQ) | 125 | 0,00 | 9,83 | 1,80 | 1,63 | 1,75 | 5,41 |
| ICAR16 total score | 178 | 5,00 | 16,00 | 12,62 | 2,49 | -0,65 | 0,00 |
| Groningen mean total score | 192 | 0,14 | 12,00 | 4,12 | 1,94 | 0,82 | 1,25 |
| EHI laterallity quotient | 192 | -100,00 | 100,00 | 72,17 | 53,21 | -2,49 | 4,97 |
| PHQ total score | 194 | 0,00 | 24,00 | 6,19 | 4,76 | 1,32 | 2,02 |
| TAS total score | 191 | 22,00 | 80,00 | 44,98 | 11,53 | 0,41 | -0,20 |
| AIS total score | 197 | 0,00 | 18,00 | 4,92 | 3,52 | 0,98 | 0,63 |
| DES total score | 188 | 29,00 | 174,00 | 68,59 | 28,77 | 1,13 | 1,31 |
| Infrequency (ZKPQ) | 194 | 0,00 | 6,00 | 2,25 | 1,45 | 0,38 | -0,35 |
| Sociability (ZKPQ) | 192 | 2,00 | 16,00 | 7,68 | 3,70 | 0,15 | -0,89 |
| Activity (ZKPQ) | 192 | 1,00 | 16,00 | 8,96 | 3,29 | -0,12 | -0,59 |

(*Continued*)

**Table 6.** (Continued)

| Key Variables in BSETS | N | Minimum | Maximum | Mean | SD | Skewness | Kurtosis |
|---|---|---|---|---|---|---|---|
| Aggression/hostility (ZKPQ) | 194 | 1,00 | 15,00 | 7,50 | 3,14 | 0,16 | -0,78 |
| Neuroticism/anxiety (ZKPQ) | 188 | 0,00 | 19,00 | 8,94 | 4,63 | 0,02 | -1,05 |
| Impulsivity (ZKPQ) | 191 | 0,00 | 8,00 | 2,42 | 2,09 | 0,71 | -0,40 |
| Sensation seeking (ZKPQ) | 194 | 0,00 | 10,00 | 4,57 | 2,59 | 0,09 | -0,90 |
| Extraversion (BFI) | 193 | 14,00 | 39,00 | 27,05 | 5,65 | -0,19 | -0,44 |
| Agreeableness (BFI) | 195 | 17,00 | 44,00 | 32,72 | 5,14 | -0,31 | -0,23 |
| Conscientiousness (BFI) | 195 | 17,00 | 44,00 | 31,94 | 5,91 | -0,22 | -0,45 |
| Emotional instability (BFI) | 194 | 10,00 | 38,00 | 23,91 | 5,69 | -0,19 | -0,14 |
| Openness (BFI) | 194 | 22,00 | 49,00 | 37,46 | 5,62 | -0,51 | -0,20 |
| Naps (experiences) | 189 | 0,00 | 1,00 | 0,18 | 0,21 | 1,24 | 1,06 |
| Nap duration (experiences) | 189 | 0,00 | 120,00 | 13,36 | 20,88 | 2,22 | 5,64 |
| Company of others (experiences) | 192 | 0,00 | 1,00 | 0,80 | 0,20 | -1,14 | 1,23 |
| School (experiences) | 192 | 0,00 | 1,00 | 0,28 | 0,29 | 0,52 | -1,17 |
| Work (experiences) | 192 | 0,00 | 1,00 | 0,24 | 0,30 | 0,79 | -0,94 |
| Alcohol (experiences) | 192 | 0,00 | 1,00 | 0,18 | 0,22 | 1,30 | 1,46 |
| Movie (experiences) | 192 | 0,00 | 1,00 | 0,40 | 0,30 | 0,28 | -0,97 |
| Computer (experiences) | 192 | 0,00 | 1,00 | 0,66 | 0,31 | -0,75 | -0,60 |
| Smart device (experiences) | 192 | 0,00 | 1,00 | 0,63 | 0,37 | -0,19 | -0,52 |
| Time on smart device (experiences) | 184 | 0,00 | 529,14 | 64,79 | 67,69 | 2,56 | 11,88 |
| Sex (experiences) | 189 | 0,00 | 0,86 | 0,12 | 0,19 | 1,69 | 2,49 |
| Car driving (experiences) | 190 | 0,00 | 1,00 | 0,17 | 0,26 | 1,65 | 1,82 |
| Public transportation (experiences) | 190 | 0,00 | 1,00 | 0,37 | 0,30 | 0,24 | -1,29 |
| Conflict (experiences) | 190 | 0,00 | 0,85 | 0,07 | 0,13 | 2,28 | 7,07 |
| Self-reported sleep onset (minutes) | 192 | 3,29 | 71,67 | 17,74 | 11,42 | 1,98 | 5,26 |
| Dream reported (experiences) | 192 | 0,00 | 1,00 | 0,27 | 0,28 | 1,25 | 1,99 |
| Happiness (dream) | 130 | 1,00 | 5,00 | 2,50 | 1,09 | 0,27 | -0,71 |
| Guilt (dream) | 130 | 0,00 | 4,00 | 1,61 | 0,90 | 1,32 | 0,98 |
| Satisfaction (dream) | 130 | 1,00 | 5,00 | 2,24 | 1,07 | 0,72 | -0,20 |
| Surprise (dream) | 130 | 1,00 | 5,00 | 2,70 | 1,11 | -0,03 | -0,89 |
| Fear (dream) | 130 | 1,00 | 5,00 | 2,22 | 1,15 | 0,74 | -0,37 |
| Curiosity (dream) | 130 | 1,00 | 5,00 | 2,66 | 1,09 | 0,09 | -0,74 |
| Sadness (dream) | 129 | 1,00 | 5,00 | 2,14 | 1,08 | 0,76 | -0,36 |
| Security (dream) | 129 | 1,00 | 5,00 | 2,15 | 1,02 | 0,76 | -0,03 |
| Anger (dream) | 130 | 1,00 | 5,00 | 1,85 | 1,00 | 1,46 | 2,01 |
| Shame (dream) | 130 | 1,00 | 5,00 | 1,56 | 0,86 | 1,98 | 3,83 |
| Uncertainty (dream) | 131 | 1,00 | 5,00 | 2,78 | 1,28 | 0,05 | -1,10 |

For EEG-derived sleep macrostructure, duration variables are in minutes. For MCTQ-derived variables, durations are reported in hours, and time points are reported in hours relative to midnight. For experiences, we report relative frequencies (proportion of days with this experience reported relative to total days in study). For questionnaires, we report raw scores.

In sum, while BSETS is not designed to be a population-representative sample, we found that participants are reasonably demographically diverse. Although most of our participants are young, and well-educated, participants with other demographic characteristics are also represented. We found high compliance with our protocol, with only 4% of participants dropping out of the study completely and over 95% of the nights from active participants providing diary and EEG data. The relatively young age of participants may have been a factor

supporting high compliance, and in elderly or pathological population samples, which are more commonly studied in the sleep research field, may have different habits or engagement styles with respect to smartphone-activated technology or worn devices, and potential compliance issues arising from this should be considered in future research.

## Quantitative EEG measures

Dreem 2 devices record EEG from five channels with various desirable and undesirable characteristics due to the topography of their references. Channels with more distant fronto-occipital references more closely approximate standard monopolar laboratory PSG, but are likely more contaminated by noise due to the less robust occipital electrodes of Dreem 2. Fronto-frontal channels are less likely to accurately measure topographically widespread activity such as sleep spindles [66, 67], and especially also slow waves [68], but they are also less likely to be contaminated by noise (**Fig 2**). Various definitions of the automatically detected artifacts are possible (see Methods), which may eliminate some of the problems resulting from the presence of noise. We inspected the mean PSDs from multiple channels and artifact detection methods in order to examine which method of quantitative EEG registration has the most favorable characteristics.

First, we analyzed data availability (loss of channels/nights due to artifacts) and data quality (visual inspection of spectra and the ability to reproduce their known correlations with age) using four artifact definition methods. The results are summarized in **Table 7**.

We found that data availability is excellent on fronto-frontal channels, most prominently the F7-F8 channel, wherein over 80% of available nights provided a PSD estimate even with the most stringent artifact definition. However, data availability was significantly reduced on fronto-occipital channels (F7-O1, F8-O2), where more stringent methods excluded over half of nights as too noisy to provide a PSD estimate. Most of the channel quality (proportion of non-artifactual epochs) variance (56.7%-79.7%) was accounted for by participant identity, suggesting that some participants may have been more conscientious in their use of the Dreem 2 EEG devices. Conversely, because multiple recordings were available for most participants, the measure of average PSD could be calculated for a much higher proportion of participants than from individual nights.

On **Fig 3**, we illustrate findings from one definition which discards all epochs with a >25% artifact probability and all channels with an overall quality of <20%. We highlight PSDs by night and by participant on a channel with high data availability, F8-F7. Overall, we found that both night- and individual-level average PSD estimates resemble those obtained using standard PSG, with typical delta and sigma peaks. As expected given the frontal topography of key electrodes, sigma peak frequency was low at ∼ 12 Hz, possibly indicating slow rather than fast spindles. A minority of nights (and, due to the high interindividual similarity of channel quality, participants) still may be affected by artifacts, which most frequently contaminated activities above the sigma range. As we anticipated, fronto-frontal channels had lower voltage than fronto-occipital channels, but even these (with the exception of Fp1-F7) retained characteristic spindle peaks. The negative correlation between PSD, and NREM delta and sigma activity, was present on all channels, but most prominent on fronto-occipital channels, especially in the delta range. Similar visualizations are provided in the **Supplementary data** for other artifact definitions.

Quantitative EEG analyses in BSETS are limited by the frontal topography of available EEG channels. We quantitively assessed the scope of this limitation by comparing PSD estimates from frontal and other channels in an external dataset with a full PSG cap (**S1 Text**). We found that between-participant correlations of PSD estimates between frontal and other channels are

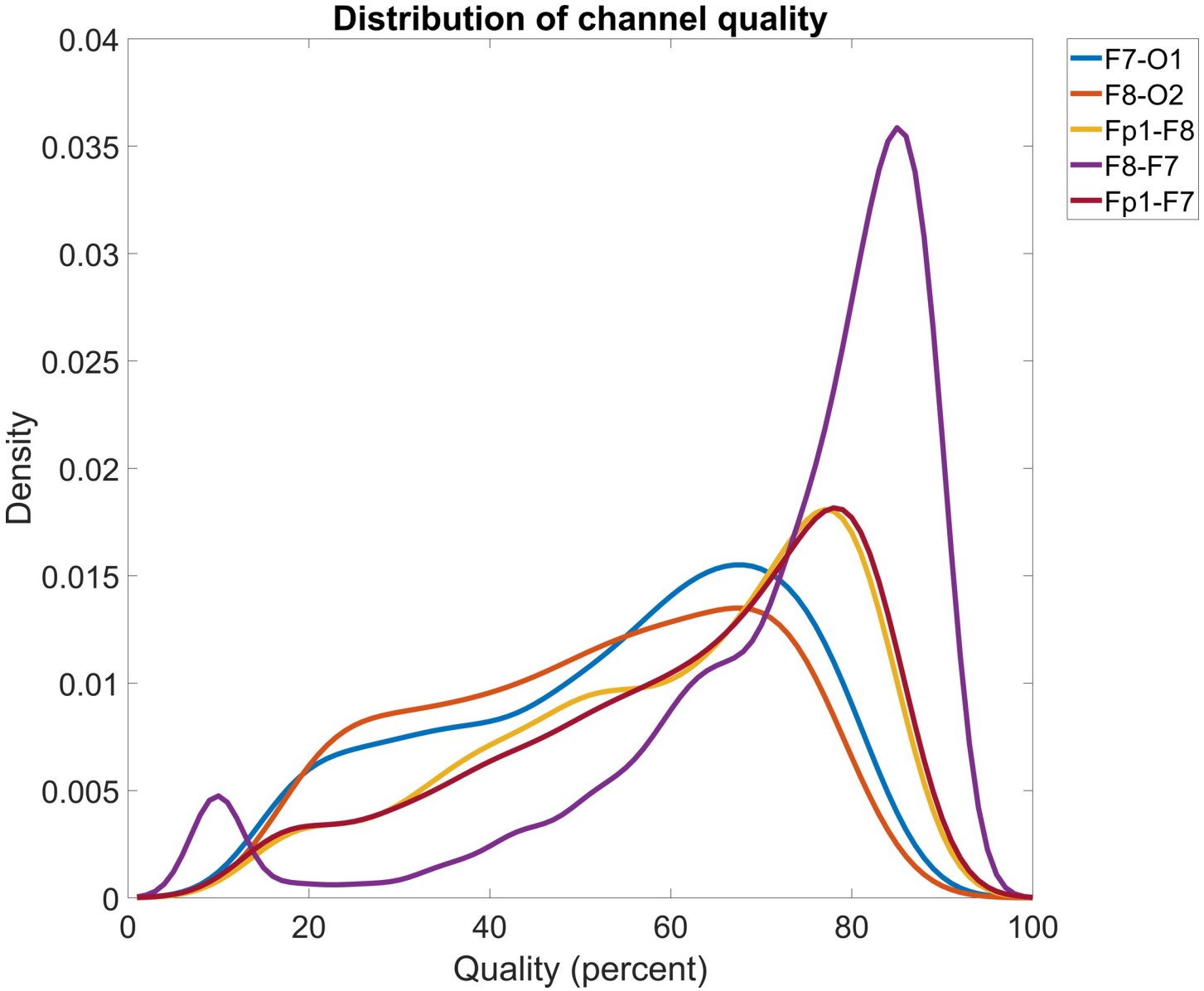

**Fig 2. The distribution of channel quality (proportion of epochs with <50% artifact probability).**

reasonably high (typically r = 0.7–0.8). A local minimum is observed in the fast spindle frequency range, but even here a minimal correlation of >0.5 is observed. This suggests that the frontal PSD estimates in BSETS are reasonable proxies for activities which occur without a characteristic frontal topography.

In sum, our preliminary analyses found that the quality of EEG data in BSETS is sufficient for qEEG analysis. We tested the feasibility of this approach by calculating a simple and well-known EEG metric, PSD, and investigate if findings in the literature can be replicated using BSETS EEG data. While artifacts contaminated all channels, especially fronto-occipitally ones, this problem could be mitigated using a complimentary automatic artifact detection algorithm, which performed reasonably well regardless of settings. The majority of data from fronto-frontal and a large minority from fronto-occipital channels could be recovered even on a night-by-

**Table 7. The effect of various artifact definitions on data availability and the reproducibility of the age-PSD correlation.**

|  | Channel | F7-O1 | F8-O2 | Fp1-F8 | F8-F7 | Fp1-F7 |
|---|---|---|---|---|---|---|
| QE>75%, QC>20% | *Valid N (nights)* | 1115 | 1077 | 1215 | 1251 | 1204 |
|  | *Valid N (participants)* | 189 | 189 | 192 | 191 | 192 |
|  | *NREM Delta-age correlation* | -0,46 | -0,46 | -0,27 | -0,34 | -0,12 |
|  | *NREM Sigma-age correlation* | -0,10 | -0,11 | -0,02 | -0,09 | 0,04 |
| QE>75%, QC>20% | *Valid N (nights)* | 1115 | 1078 | 1215 | 1251 | 1204 |
|  | *Valid N (participants)* | 189 | 189 | 192 | 191 | 192 |
|  | *NREM Delta-age correlation* | -0,47 | -0,47 | -0,26 | -0,35 | -0,11 |
|  | *NREM Sigma-age correlation* | -0,09 | -0,10 | 0,00 | -0,10 | 0,05 |
| QE>75%, QC>50% | *Valid N (nights)* | 765 | 691 | 933 | 1160 | 958 |
|  | *Valid N (participants)* | 164 | 158 | 181 | 190 | 178 |
|  | *NREM Delta-age correlation* | -0,48 | -0,41 | -0,33 | -0,34 | -0,20 |
|  | *NREM Sigma-age correlation* | -0,13 | -0,09 | -0,07 | -0,10 | 0,02 |
| QE>50%, QC>50% | *Valid N (nights)* | 765 | 691 | 933 | 1160 | 958 |
|  | *Valid N (participants)* | 164 | 158 | 181 | 190 | 178 |
|  | *NREM Delta-age correlation* | -0,50 | -0,44 | -0,33 | -0,35 | -0,19 |
|  | *NREM Sigma-age correlation* | -0,12 | -0,09 | -0,05 | -0,11 | 0,03 |

We show outcomes for two definitions of minimum epoch quality (QE, set at 50% and 75%) and two definitions of minimum channel quality (QC, set at 20% and 50%, in both cases referring to the minimum proportion of epoch that had to exceed a non-artifact probability of 50%). For each definition, we show for each channel the number of nights with available data, the number of participants with at least one night of available data, and the correlation of NREM absolute delta (0.5–4 Hz) and low sigma (10–13 Hz) power with age. Night and participant numbers are shown for NREM, in some cases these are slightly different in REM (see Supplementary data).

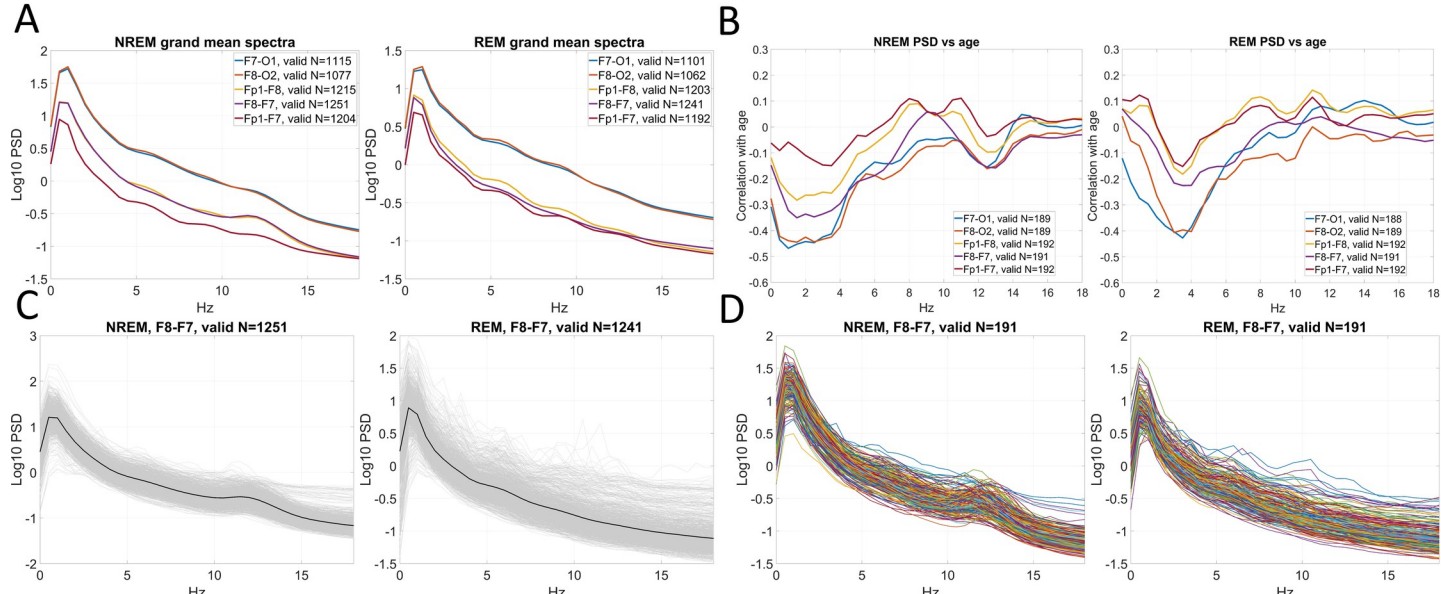

**Fig 3. An illustration of qEEG data in BSETS. Panel A**: grand average PSDs calculated across all available nights, displayed by channel. Note the higher voltage of fronto-occipital channels and the relative absence of spindle peaks from the Fp1-F7 channel with a close bipolar reference. **Panel B**: the correlation between individual average PSD (in 0.5 Hz bins) and age. **Panel C**: average PSD estimates of all nights from F8-F7 (thin grey lines), overlain with the grand average (thick black line). **Panel D**: individual average PSD estimates (averages across all nights from the same participant) from F8-F7. All data is shown after the exclusion of channels with <20% data quality and epochs with >25% artifact probability.

night basis. The resulting PSD values exhibited typical delta and sigma peaks and the expected correlations with age. Encouragingly, these findings were present even for the fronto-frontal F8-F7 channel which has high data availability. While our goal was to demonstrate how a fully automatic pipeline can process BSETS EEG data, we note that an additional visual inspection to eliminate nights with abnormal EEG spectra could further improve data quality.

An additional benefit of our use of mobile EEG devices [69] over laboratory PSG is the ease of protocol adherence and ecological validity, as participants can record EEG in their homes. While laboratory recordings are often affected by first night effects [70], recent research has shown that there is relatively little difference between EEG recordings undertaken in the laboratory or at home [71].

## Examples of research opportunities

BSETS can be used to investigate a large number of hypotheses related to the relationship between sleep, normal and pathological human traits, and daily experiences. Here we enumerate a number of currently unanswered research questions which can be readily clarified using BSETS data.

1. It is known from some studies that at least moderate associations exist between self-reported circadian timing and objective (usually actigraphic) measures of actual sleep and wake timing [72–74]. However, to our knowledge no study compared to self-reported and EEG-measured patterns of the timing of daily activity.

2. Self-rated sleep quality has low but significant correlations with PSG-rated sleep structure [25]. Data from this question, however, comes from between-person studies. It is unknown if and to what extent within-person variation in subjective sleep quality is related to within-subject variation in objective sleep quality - in other words, if objectively better sleep on a night leads to higher subjective ratings by the same person.

3. There is widespread belief that sexual intercourse improves sleep, with some empirical comparison in a recent diary-based study with subjective sleep reports [23]. It is unknown if sexual intercourse also improves objectively measured sleep. Similar replications are warranted for all previous findings relying exclusively on self-reports - in other words, it is important if a daily experience affects objectively measured sleep or only the subjective perception of it.

4. There are numerous algorithms developed to detect morphological features, such as sleep spindles, in laboratory EEG data [75, 76]. While these algorithms may not be immediately applicable to mobile EEG data, after the development and validation of new algorithms morphological features can be detected and used as dependent variables in BSETS EEG data.

At the Editor's suggestion, we include a proof of principle analysis investigating the effects of exercise on sleep. A recent daily observational study [77], similar in design to BSETS, found increased REM latency, reduced REM and slightly increased N3 after mild, but not after vigorous exercise in the elderly. A recent meta-analysis [16], following previous similar studies [18, 78, 79] highlights that current recommendations advise against exercise in sleep, and quantitively synthesizes 23 investigating the effect of evening exercise on sleep. It revealed borderline significant increases in REM latency and SWS, and a highly significant decrease in N1. However, the largest study in the meta-analysis had only 24 participants, and the total number of across all studies in the meta-analysis was approximately equal to the intended final size of BSETS (N = 266). We attempted to replicate these findings using a within-participant design

with multilevel regression models, using EEG-derived sleep structure as the dependent variable, and the exercise and a random intercept per participant as independent variables, implemented in the lmer() function in the lmerTest R package, which is similar to lme4 but provides p-values from multilevel models using Satterthwaite's approximation. Due to violations of the homoskedasticity and normality of residuals assumptions (see also **Table 6**) we log-transformed WASO before model fitting and for sleep onset latency and REM latency we specified a gamma distribution with a log link function, implemented with the glmer() function. We controlled all models for day of the week (specified as weekend or weekday). We found that exercise slightly but significantly (p = 0.006) increases sleep onset latency, but not other sleep macrostructure variables (**Table 8**).

Our findings do not directly replicate the results of the previous meta-analysis or a recent daily observational study [77], but suggests that exercise may complicate the onset of sleep. The limitations of our analysis include that we did not control for the time of exercise relative to sleep, and thus could not distinguish between exercise any time during the day and specifically in the evening.

BSETS is designed to be an accessible database which can be used and re-used by both academic and independent researchers for a plethora of purposes, the most important among them being to further explore relationships between what we do, how we sleep, and whether that might change depending on who or what we are, psychologically. Our hope is that this dataset will empower scientists to address longstanding lacunae in sleep literature, and to enhance our understanding of what influences sleep, what sleep influences, and how. BSETS is projected to finalize data collection and cleaning by Summer 2023. Once data collection and preprocessing is completed by late 2023, the BSETS research is open for requests to release data or collaborate with others interested in our resources.

Participants in BSETS are mainly young and healthy. Nevertheless, we envision BSETS findings to have significant clinical applications for the elderly and for those with sleep disorders, for example, Insomnia disorder. This is because the substantial statistical power of BSETS can be leveraged to identify DAEs which are associated with changes in subsequent sleep quality and if such DAEs are found, they can be used to design evidence-based interventions in at-risk samples. For example, if certain dietary patterns are associated with improved sleep in BSETS, at-risk individuals can be encouraged to follow this dietary pattern to improve

**Table 8. The effects of participant-reported exercise on subsequent sleep macrostructure.**

| Selected key variables | B | T | p |
|---|---|---|---|
| Sleep onset latency | 1.123** | 2.747 | 0.006 |
| REM onset latency | 1.025 | 0.816 | 0.415 |
| Wake after sleep onset | 0.918 | -1.51 | 0.131 |
| Sleep efficiency | -0.014 | -0.527 | 0.98 |
| Total sleep time | -4.038 | -0.701 | 0.483 |
| N1 duration | -0.636 | -1.140 | 0.254 |
| N2 duration | -1.500 | -0.421 | 0.674 |
| SWS duration | -0.823 | -0.526 | 0.599 |
| REM duration | -2.455 | -1.033 | 0.302 |

All variables are regression coefficients (after appropriate transformations in case of nonlinear models) from multilevel models. Estimates from log-transformed models are exponentiated. All models are controlled for day of week.

** p<0.01.

their sleep as well. Importantly, because such an intervention is based on an explicit hypothesis in contrast to the exploratory BSETS, a smaller sample can enough to confirm efficacy. In these smaller samples, the use of standard laboratory PSG may be feasible if, for example, an elderly population has technical issues with the use of smartphone-based mobile EEG technology in their homes.

## Limitations

We designed BSETS to address several common issues affecting research into bidirectional relationships between daily experiences and sleep, most importantly issues concerning statistical power and the objective measurement of sleep. Nevertheless, BSETS has its own limitations which we discuss here.

During recruitment to BSETS, we prioritized sample size and high adherence to demographic representativity. As a result, BSETS participants are mainly healthy young participants, and its main goal is to study the effects of common daily experiences on normal sleep. BSETS findings may not automatically generalize to an elderly population or to those with sleep disorders. However, this limitation can be overcome in BSETS-based studies by using between-individual variation as a moderator. In other words, we can investigate if a certain relationship between DAEs and sleep is different as a function of demographic (e.g. age and sex) or health-related (e.g. trait depression, overall sleep quality) characteristics. Furthermore, BSETS data can be used to identify DAEs with effects on normal sleep, which can then be used in targeted interventions in more focused studies of the elderly or those with sleep disorders.

Our mobile EEG devices enable an accurate automatic extraction of hypnograms [63], and, as our own analyses presented here show, provide adequate recording quality to be recordings for at least basic quantitative EEG analyses. However, these devices only contain frontal and occipital electrodes, resulting in a limited topographical mapping of EEG activities. Most significantly, the lack of central channels affects our ability to detect fast spindles, although quantative analyses (**S1 Text**) suggest that frontal PSD estimates may be reasonable proxies for activity with other topographies. Furthermore, signal quality may not be sufficient for certain quantitative analyses, and there is overall little literature about quantitative EEG analyses with mobile dry EEG devices.

An important conceptual limitation comes from the fact that by observing the same individuals over multiple days in BSETS, it is only daily experiences with non-zero within-individual variance that is informative. Simply put, if participants experience very similar experiences and very similar sleep on all days, we can learn little about the relationship between these two variables in a daily observational study like BSETS, which effectively uses participants as their own controls. However, previous studies [20, 22, 24] reported moderate intraclass correlation coefficients, that is, indicators of within-individual similarity, of daily experiences and sleep variables, suggesting that in practice, we will see sufficient within-individual variance in BSETS to enable meaningful analyses.

As we aimed at a broad sampling of traits and daily experiences in our questionnaire-based measurements in BSETS, in some cases we were limited by practical constraints about the amount of information we could obtain using questionnaires about specific traits/experiences. For example, most daily experiences are only reported as either having or not having occurred, with no additional information such as for example, the specific time of day an event occurred, which may affect its ability to affect sleep. Our participants did not recruit caffeine intake, which we excluded due to the existence of a high-quality literature (see e.g. [15, 80]) on the effects of caffeine on sleep.

Finally, as a general limitation, while BSETS contains information about a wide variety of traits, experiences and sleep variables, there is a practically infinite number of possible relationships of interest between these constructs. Therefore, there will always be important research hypotheses about the relationship between daily experiences, traits and sleep which cannot be addressed with BSETS data.

## Conclusions

How daily experiences influence sleep while themselves depending upon it has been a focus of interest since even before the birth of modern sleep research, but comparatively few studies have been published at least until the nineties [13]. There has been growing interest in this field recently, with many new studies using advanced methods studying the relationship between sleep and, for instance, diet [14], exercise [16–18], mood [19], stress [20–22] or sexual activity [23]. These studies expanded knowledge about this field considerably.

However, some outstanding issues remain. First, while some studies [22, 24] used EEG to provide an objective measurement of sleep, this is not standard procedure and this may be problematic as self-reports of sleep are only moderately accurate [25]. To our knowledge, no daily observational study used quantitative EEG to establish the link between daily experience and sleep, although based on experimental studies [81, 82] it is highly expected that daily experiences will affect waveforms during the subsequent nights, for example because richer experiences require additional synaptic downscaling through increased slow-wave activity [4]. Second, most datasets are aimed to study the relationship between sleep and at most a handful of daily experiences. Third, while many datasets are large, sometimes with hundreds of participants and thousands of nights [20, 23], sample sizes in this field are variable [19] which may limit statistical power to detect small but genuine associations.

We aim to bridge these gaps with BSETS by establishing a daily observational dataset which is 1) large in size, 2) contains both objective and subjective ratings of sleep and 3) contains data about a large array of daily experiences, including social life, activities, food and alcohol consumption and digital device use. By providing a basis for sleep researchers to investigate, evaluate, and quantify bidirectional associations between both objective (EEG) and subjective measures of sleep, against DAEs, BSETS provides the opportunity to embed existing sleep studies in the background of a cohesive, broad-spectrum framework of how DAE and sleep features relate to one another, subjectively and objectively as recorded on the EEG.

## Supporting information

**S1 Text. Description of the similarity of PSD estimates from frontal and non-frontal channels.**
(DOCX)

## Acknowledgments

The authors express their gratitude to research assistants and contributing undergraduate students Réka Finta, Pardis Adibi, Eliya San and Sára Wolf for their outstanding assistance in data collection and preparation. We would like to thank the teaching staff (in particular, our main contact person, course administrator Edit Czeglédi) of the Institute of Behavioural Sciences at Semmelweis University for their support in recruitment and data collection. We thank Róbert Bódizs, Péter Simor and Orsolya Szalárdy for helpful comments during the preparation of the BSETS study plan and for lending us EEG devices. Furthermore, the authors express their gratitude to everyone who participated in the BSETS protocol.

## Author Contributions

**Conceptualization:** Wael Taji, Péter Przemyslaw Ujma.

**Data curation:** Wael Taji, Róbert Pierson, Péter Przemyslaw Ujma.

**Formal analysis:** Wael Taji, Péter Przemyslaw Ujma.

**Funding acquisition:** Péter Przemyslaw Ujma.

**Investigation:** Péter Przemyslaw Ujma.

**Methodology:** Péter Przemyslaw Ujma.

**Project administration:** Róbert Pierson, Péter Przemyslaw Ujma.

**Resources:** Péter Przemyslaw Ujma.

**Software:** Péter Przemyslaw Ujma.

**Supervision:** Péter Przemyslaw Ujma.

**Validation:** Péter Przemyslaw Ujma.

**Visualization:** Péter Przemyslaw Ujma.

**Writing – original draft:** Wael Taji, Róbert Pierson, Péter Przemyslaw Ujma.

**Writing – review & editing:** Wael Taji, Róbert Pierson, Péter Przemyslaw Ujma.

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
