## [Decision Letter · Decision Letter 0]

17 May 2023

PONE-D-23-10162Protocol of the Budapest Sleep, Experiences, and Traits Study: an accessible resource for understanding associations between daily experiences, individual differences, and objectively measured sleepPLOS ONE

Dear Dr. Péter,

Thank you for submitting your manuscript to PLOS ONE. After careful consideration, we feel that it has merit but does not fully meet PLOS ONE’s publication criteria as it currently stands. Therefore, we invite you to submit a revised version of the manuscript that addresses the points raised during the review process.

The article has been evaluated by two expert reviewers. Both believe that the proposed dataset may be of great interest to the research community.

However, they are concerned about the bias of the dataset towards young, healthy university students.

This may produce a significant bias in the data analysis and experience results. Please respond carefully to their comments and concerns, answering each comment and including in the text the missing information.

We look forward to receiving your revised manuscript.

Kind regards,

Luigi Borzì, Ph.D.

Academic Editor

PLOS ONE

Journal Requirements:

"This publication has been supported by the National Research, Development and Innovation Fund of the Ministry of Innovation and Technology (grant ID: OTKA PD 138935).."

"PPU was supported by grant OTKA PD 138935, received from the National Research, Development and Innovation Fund of the Ministry of Innovation and Technology (https://nkfih.gov.hu/). The funders did not and will not have a role in study design, data collection and analysis, decision to publish, or preparation of the manuscript."

4. We note that Figure 1 in your submission contain copyrighted images. All PLOS content is published under the Creative Commons Attribution License (CC BY 4.0), which means that the manuscript, images, and Supporting Information files will be freely available online, and any third party is permitted to access, download, copy, distribute, and use these materials in any way, even commercially, with proper attribution. For more information, see our copyright guidelines: http://journals.plos.org/plosone/s/licenses-and-copyright.

(1) You may seek permission from the original copyright holder of Figure 1 to publish the content specifically under the CC BY 4.0 license. 

**Additional Editor Comments:**

From the editor's perspective, I suggest including a proof-of-principle analysis using at least one of the examples of research opportunities described in section 4.2. This can be of great use in demonstrating the enormous potential of the collected dataset.

Moreover, I recommend that you carefully review the manuscript to improve readability.

- Please avoid the excessive use of parentheses, as they reduce the overall reading flow. Instead, whenever possible, spell out the content of the text.

- Check carefully for typos and errors. For example, a parenthesis is missing in "(i.e. media and private companies.)", while parentheses are duplicated in "(i.e. media and private companies.)".

- Define acronyms the first time they are used. Then, use the acronyms. For example, the definition of AED is duplicated in the introduction, while PSG, REM, EEG, and IRB are not defined.

- Avoid excessive use of parentheses. Instead, explain the content in the text.

- It is worth reporting the total number of participants with a complete data set (good quality EEG, completed sleep diary and completed daily experience diary) for 7 consecutive nights.

- Check that all labels and legends are visible in the figures. In Figure 3 they are not.

Reviewers' comments:

Reviewer's Responses to Questions

**Comments to the Author**

1. Does the manuscript provide a valid rationale for the proposed study, with clearly identified and justified research questions?

Reviewer #1: Partly

Reviewer #2: Yes

2. Is the protocol technically sound and planned in a manner that will lead to a meaningful outcome and allow testing the stated hypotheses?

Reviewer #1: Partly

Reviewer #2: Partly

3. Is the methodology feasible and described in sufficient detail to allow the work to be replicable?

Reviewer #1: Yes

Reviewer #2: Yes

4. Have the authors described where all data underlying the findings will be made available when the study is complete?

Reviewer #1: Yes

Reviewer #2: Yes

5. Is the manuscript presented in an intelligible fashion and written in standard English?

Reviewer #1: Yes

Reviewer #2: Yes

6. Review Comments to the Author

You may also provide optional suggestions and comments to authors that they might find helpful in planning their study.

Reviewer #1: This study presents an ecological dataset focusing on sleep, daily activities/experiences, and psychophysiological measures, with the aim of investigating multidirectional relationships between these variables. A total of 208 healthy subjects were enrolled and longitudinally followed for seven days, with semiquantitative and subjective measures of psychological traits, sleep quality, and daily activities/experiences collected each day using standardized tests, diaries, and questionnaires. Objective measures were also gathered through a portable EEG device that was previously tested, allowing for overnight monitoring of sleep. The authors discussed the demographic features of participants and addressed technical challenges associated with portable EEG recordings to obtain objective data on sleep. Finally, they discussed the potential applications of the proposed dataset for future research, highlighting the importance of investigating the relationships between sleep, daily activities and experiences, and psychophysiological traits.

Overall, the study provides a valuable dataset that can be particularly useful for future research. However, I have some major concerns to be addressed:

- The participants in the study are overwhelmingly young, with the majority being under 25 years old. Therefore, the study cohort is not representative of the population in which specific sleep disorders are more commonly present. This is a relevant limitation that should be better examined.

- The enrollment of a large number of university students as the majority of the study population, as well as the involvement of individuals belonging to the same family unit, may represent a significant bias in the study due to the sharing of demographic characteristics and lifestyle habits, possibly affecting experimental findings.

- The results regarding the measurements of psychological traits, subjective measures of sleep, and daily activities/experiences are completely absent in the text.

- The manuscript would benefit from concise writing, as it can be verbose and occasionally repetitive. For instance, the first paragraph of the discussion reiterates background information that was already introduced in the introduction. Additionally, although lengthy, the introduction does not provide a state-of-the-art overview of the main topic of the study, namely the relationships between sleep, psychological traits, and daily activities, but is rather focused on numerous methodological details.

- The study abstract and introduction should clearly state that the purpose of the study is to present a dataset and not to investigate the relationship between the variables included in the dataset (sleep, psychological traits, and daily activities).

- The high compliance of study participants can be explained by the involvement of young individuals who are familiar with the use of technological instrumentation. The approach used in this study may not be easily applicable to a population of elderly subjects. Please discuss.

- The authors should expand the discussion of the data. In particular, I suggest considering the application of the proposed methodological approach in specific pathological conditions as a possible perspective.

Reviewer #2: The Authors present an experimental protocol specifically design to construct a complete dataset which would allow for the investigation of daily experiences, psychological traits and their relation to quantitative (and qualitative) sleep metrics - the latter retrieved from qEEG and subjective assessments.

The aim of the study is clearly stated and the paper is well-organised.

However, I have a few remarks regarding the methodology, which I believe could be addressed during the manuscript revision; you may find them in the following.

(1) As you stated, the Data Acquisition process is still ongoing and will be complete within the next few months. As pointed out in paragraph 2.1, the sample population includes mainly young participants. Different age ranges result in various daily-activity habits and experiences, and may result in different quantitative sleep metrics or sleep fragmentation patterns. Do you plan on extending the dataset by including a higher number of participants of different/higher age? If not, could you please discuss the limitations that this population bias may lead to?

(2) The presented questionnaires (Sect. 2.3.2) are well designed. However, did you investigate the amount of caffeine intake, the presence of medication of any kind (targeting cardiovascular or breathing disorders, neurological disorders, psychological disorders) or the presence of familiarity with any neurological disorder? Cardiovascular impairment or neurological disorders may significantly impact the sleep pattern, resulting in higher presence of spindles, arousals or high sleep fragmentation. Please discuss.

(3) In Section 2.3.3. the PSD is introduced as elected quantitative metric for the evaluation of qEEG. Later on, in Section 4.3, you mention that the lack of central channels may lead to an underestimation of the presence/number of sleep spindles (and sigma rhythm). In view of this, does the PSD still remain a reliable metric, or there may be the need for including other quantitative assessments retrieved from the EEG? Indeed, morphological (i.e., time-domain) characteristics are also widely investigated in literature. I believe this is a key point to be discussed.

I hope these remarks may be positively addressed in the revised version of the manuscript.

7. PLOS authors have the option to publish the peer review history of their article (what does this mean?). If published, this will include your full peer review and any attached files.

Reviewer #1: No

Reviewer #2: No

---

## [Author Response · Author response to Decision Letter 0]

23 Jun 2023

We thank the Reviewers for their comments on our manuscript. In accordance with their recommendations, we thoroughly re-wrote the paper. We shortened and simplified the Introduction and the Discussion, removed duplications and included more details about the findings of previous literature. At the Editor’s suggestion, we included a proof of concept analysis about the effect of exercise on sleep. We found that exercise increases sleep onset latency, but does not affect other sleep variables.

Since the paper was submitted, we received communication from Dreem, our mobile EEG headband’s manufacturer, that the channel names Fp1-O2 and Fp2-O1 were included by their error and in fact these are F7-O2 and F8-O1, respectively. As these are redundant to our F7-O1 and F8-O2 channels, we removed these channels from all analyses, which now only discuss five EEG channels.

2. Thank you for stating the following in the Acknowledgments Section of your manuscript: "This publication has been supported by the National Research, Development and Innovation Fund of the Ministry of Innovation and Technology (grant ID: OTKA PD 138935).." We note that you have provided funding information that is not currently declared in your Funding Statement. However, funding information should not appear in the Acknowledgments section or other areas of your manuscript. We will only publish funding information present in the Funding Statement section of the online submission form. Please remove any funding-related text from the manuscript and let us know how you would like to update your Funding Statement. Currently, your Funding Statement reads as follows: "PPU was supported by grant OTKA PD 138935, received from the National Research, Development and Innovation Fund of the Ministry of Innovation and Technology (https://nkfih.gov.hu/). The funders did not and will not have a role in study design, data collection and analysis, decision to publish, or preparation of the manuscript." Please include your amended statements within your cover letter; we will change the online submission form on your behalf.

We deleted the funding-related text from Acknowledgements. We would like to leave the funding statement unaltered as it already contains the funding sources in full, the text in the Acknowledgements was a repetition of this.

We believe that this information is not correct. We already posted a repository link to our database as it currently stands. However, as data collection is still in progress (as required by PLoS One protocol policies) we could not already post the full database. We would like to leave the Data Availability statement unaltered, give readers free access to a partial database, while the full database will become available for interested researchers upon completion of the protocol.

4. We note that Figure 1 in your submission contain copyrighted images. All PLOS content is published under the Creative Commons Attribution License (CC BY 4.0), which means that the manuscript, images, and Supporting Information files will be freely available online, and any third party is permitted to access, download, copy, distribute, and use these materials in any way, even commercially, with proper attribution. For more information, see our copyright guidelines: http://journals.plos.org/plosone/s/licenses-and-copyright.

Figure contains images we specifically requested from Dreem Inc for publication in this paper and as such they are not protected by copyright. We uploaded our email exchange with their customer service as a supplementary file.

Additional Editor Comments:

From the editor's perspective, I suggest including a proof-of-principle analysis using at least one of the examples of research opportunities described in section 4.2. This can be of great use in demonstrating the enormous potential of the collected dataset.

We highly appreciate the Editor’s suggestion. We intend to publish the research projects outlined in section 4.2 as separate papers. However, we added a proof of principle analysis about the effect of exercise on sleep. Using a within-participant design with multilevel regression models, we found that exercise increases sleep onset latency, but doesn’t affect other sleep characteristics.

- Please avoid the excessive use of parentheses, as they reduce the overall reading flow. Instead, whenever possible, spell out the content of the text.

We revised the manuscript and specifically aimed to eliminate parentheses whenever possible.

- Check carefully for typos and errors. For example, a parenthesis is missing in "(i.e. media and private companies.)", while parentheses are duplicated in "(i.e. media and private companies.)".

Reply: We have amended parenthetical omissions present in the previously submitted version of the manuscript.

- Define acronyms the first time they are used. Then, use the acronyms. For example, the definition of AED is duplicated in the introduction, while PSG, REM, EEG, and IRB are not defined.

Reply: We have responded by ensuring that acronyms are consistently used throughout the manuscript after introduction and definition with the exception of acronymic terms used in title headings or subheadings, or in direct quotation where the acronym employed in the manuscript was not used. The example given of AED (understood to mean DAE, a common term in the manuscript) was amended, and PSG, REM, EEG, and IRB were given definitions and consistently used acronymically thereafter. Common acronyms used in the abstract (i.e. EEG) were defined in the main text body and left undefined in the abstract for the sake of brevity.

- It is worth reporting the total number of participants with a complete data set (good quality EEG, completed sleep diary and completed daily experience diary) for 7 consecutive nights.

Unfortunately it is not straightforward to give a single total number of participants with such characteristics because slight variations exist within question – for example, some participants may not respond to one question within their diaries, even though the diary was returned and all other measurements completed. Total N may slightly vary as a function of the exact variable in question. However, at a Reviewer’s request, we report detailed descriptive statistics in a new Table 6. For the majority of variables (with key exceptions Chronotype, which requires no limits on free days for sleep duration; and dream-related variables, which require the participant having reported at least one dream) valid N exceeds 190.

- Check that all labels and legends are visible in the figures. In Figure 3 they are not.

We have increased the font size on Figure 3.

Reviewer #1: 

- The participants in the study are overwhelmingly young, with the majority being under 25 years old. Therefore, the study cohort is not representative of the population in which specific sleep disorders are more commonly present. This is a relevant limitation that should be better examined.

We completely agree with the Reviewer’s point. Our aim with BSETS was to maximize the size of the dataset by sampling a highly compliant population in order to improve statistical power. We believe that by identifying experiences which lead to changes in sleep in a young, healthy population we highlight possible intervention targets which can be investigated in a more focused manned in a high-risk population, for example, those with sleep disorders or the elderly. In order to clarify this, we added the following limitation to the manuscript:

“During recruitment to BSETS, we prioritized sample size and high adherence to demographic representativity. As a result, BSETS participants are mainly healthy young participants, and its main goal is to study the effects of common daily experiences on normal sleep. BSETS findings may not automatically generalize to an elderly population or to those with sleep disorders. However, this limitation can be overcome in BSETS-based studies by using between-individual variation as a moderator. In other words, we can investigate if a certain relationship between DAE and sleep is different as a function of demographic (e.g. age and sex) or health-related (e.g. trait depression, overall sleep quality) characteristics. Furthermore, BSETS data can be used to identify DAE with effects on normal sleep, which can then be used in targeted interventions in more focused studies of the elderly or those with sleep disorders.”

We also added the following text discussing the clinical significance of BSETS findings (see also some of our other responses to queries by Reviewers):

“Participants in BSETS are mainly young and healthy. Nevertheless, we envision BSETS findings to have significant clinical applications for the elderly and for those with sleep disorders. This is because the substantial statistical power of BSETS can be leveraged to identify DAE which are associated with changes in subsequent sleep quality and if such DAE are found, they can be used to design evidence-based interventions in at-risk samples. For example, if certain dietary patterns are associated with improved sleep in BSETS, at-risk individuals can be encouraged to follow this dietary pattern to improve their sleep as well. Importantly, because such an intervention is based on an explicit hypothesis in contrast to the exploratory BSETS, a smaller sample can enough to confirm efficacy. In these smaller samples, the use of standard laboratory PSG may be feasible if, for example, an elderly population has technical issues with the use of smartphone-based mobile EEG technology in their homes.”

- The enrollment of a large number of university students as the majority of the study population, as well as the involvement of individuals belonging to the same family unit, may represent a significant bias in the study due to the sharing of demographic characteristics and lifestyle habits, possibly affecting experimental findings.

We agree with the reviewer’s point on principle. However, in an empirical analysis we found that age and student status are only weakly related for most daily experiences.

In Row 2 the table below, we provide Pearson correlations between age and the frequency each of the daily experiences we sampled (total number of reported events across the 7 days). In Row 3, we report a pseudo-R expressing the relationship between student status and the same experiences. This value was calculated by calculating the between-group (students vs. non-students) variance of each experience and taking the square root of this R2 value to bring it to a scale comparable to correlations, albeit without a sign. * denotes values with p<0.05 and ** values with p<0.01.

 Nap Nap duration Car driving Time on smart device Movie Computer School Conflict Work Smart device Sports Alcohol Sex Socializing Public transport

Pearson r (age) -0,068 -0,118 ,277** -,256** -0,011 -0,040 -,514** ,222** ,444** -,251** 0,066 -0,011 -0,012 -,233** -,378**

R (student) 0,107 0.099* 0,230* 0,174 0,080 0,145* 0.471** 0,096 0.434** 0,128 0,036 0,078 0,082 0,058 0.349**

As it can be seen, younger people and students spend much less time in school and less time at work. Younger people and students also drive cars less but use public transportation and smart devices more. However, even these experiences are relatively weakly related to both age and student status, and the other experiences are unrelated to these characteristics. Thus, we believe that – with the possible exception of school and workplace attendance – most experiences sampled with relatively little bias due to the characteristics of our sample.

Concerning family units, most participants recruited a single family member. A few participants also participated themselves at a different occasion, but to our knowledge there is no case in which two participants from the same family unit participated at the same time (i.e. recorded the experiences they both participated in). We clarified this in the paper under “Study design, setting, and participant recruitment”:

“However, multiple members of the same household never participated in the study at the same time.”

- The results regarding the measurements of psychological traits, subjective measures of sleep, and daily activities/experiences are completely absent in the text.

We thank the Reviewer for pointing out this limitation. We now provide descriptive statistics for all variables in Table 6.

- The manuscript would benefit from concise writing, as it can be verbose and occasionally repetitive. For instance, the first paragraph of the discussion reiterates background information that was already introduced in the introduction. Additionally, although lengthy, the introduction does not provide a state-of-the-art overview of the main topic of the study, namely the relationships between sleep, psychological traits, and daily activities, but is rather focused on numerous methodological details.

We streamlined the manuscript to eliminate repetitions between the Introduction and Discussion and to reduce overall length. In the Introduction, we included more discussion of previous literature, and simplified the discussion of methodological issues.

- The study abstract and introduction should clearly state that the purpose of the study is to present a dataset and not to investigate the relationship between the variables included in the dataset (sleep, psychological traits, and daily activities).

We now explicitly state in the Abstract that this is a protocol paper. In the revised Introduction, we state that: 

“In the current paper, we describe the protocol of BSETS. While some empirical findings are presented in this paper, the main quantitative analyses of BSETS will follow upon completion of the dataset.”

- The high compliance of study participants can be explained by the involvement of young individuals who are familiar with the use of technological instrumentation. The approach used in this study may not be easily applicable to a population of elderly subjects. Please discuss.

We fully agree with the Reviewer’s point (see also our response to other points). BSETS was primarily designed to study the effects of common experiences on normal sleep in a healthy population, however, with the aim that the effects we identify here can be used as intervention targets for at-risk populations. It is true that we had adherence issues with the elderly due to the need to engage with technology, which is why they are not heavily represented in BSETS.

We included the following text in the Results and Discussion:

“The relatively young age of participants may have been a factor supporting high compliance, and in elderly or pathological population samples, which are more commonly studied in the sleep research field, may have different habits or engagement styles with respect to smartphone-activated technology or worn devices, and potential compliance issues arising from this should be considered in future research.”

We included the following text in the Limitations (see also our response to other queries by Reviewers):

„During recruitment to BSETS, we prioritized sample size and high adherence to demographic representativity. As a result, BSETS participants are mainly healthy young participants, and its main goal is to study the effects of common daily experiences on normal sleep. BSETS findings may not automatically generalize to an elderly population or to those with sleep disorders. However, this limitation can be overcome in BSETS-based studies by using between-individual variation as a moderator. In other words, we can investigate if a certain relationship between DAE and sleep is different as a function of demographic (e.g. age and sex) or health-related (e.g. trait depression, overall sleep quality) characteristics. Furthermore, BSETS data can be used to identify DAE with effects on normal sleep, which can then be used in targeted interventions in more focused studies of the elderly or those with sleep disorders.”

- The authors should expand the discussion of the data. In particular, I suggest considering the application of the proposed methodological approach in specific pathological conditions as a possible perspective.

We thank the Reviewer for this suggestion. In fact, one of our key goals with BSETS is to identify intervention targets for at-risk populations. We included the following text to the section “Research opportunities”:

“Participants in BSETS are mainly young and healthy. Nevertheless, we envision BSETS findings to have significant clinical applications for the elderly and for those with sleep disorders, for example, Insomnia disorder. This is because the substantial statistical power of BSETS can be leveraged to identify DAE which are associated with changes in subsequent sleep quality and if such DAE are found, they can be used to design evidence-based interventions in at-risk samples. For example, if certain dietary patterns are associated with improved sleep in BSETS, at-risk individuals can be encouraged to follow this dietary pattern to improve their sleep as well. Importantly, because such an intervention is based on an explicit hypothesis in contrast to the exploratory BSETS, a smaller sample can enough to confirm efficacy. In these smaller samples, the use of standard laboratory PSG may be feasible if, for example, an elderly population has technical issues with the use of smartphone-based mobile EEG technology in their homes.”

Reviewer #2: 

(1) As you stated, the Data Acquisition process is still ongoing and will be complete within the next few months. As pointed out in paragraph 2.1, the sample population includes mainly young participants. Different age ranges result in various daily-activity habits and experiences, and may result in different quantitative sleep metrics or sleep fragmentation patterns. Do you plan on extending the dataset by including a higher number of participants of different/higher age? If not, could you please discuss the limitations that this population bias may lead to?

We strongly agree with the Reviewers’ point. Our goal with BSETS was to prioritize statistical power by recruiting a large, highly adherent sample to study experiences which affect normal sleep. While we encouraged older participants to join (with some success as almost 20% of participants are older than 45 years old), we did not explicitly intend to recruit the elderly, mainly because using digital interfaces was a key feature of BSETS and we experienced adherence issues. However, we believe the effects we found may be used in more targeted interventions in at-risk populations. For example, if we find that certain daily experiences improve sleep in a normal population, engaging in this behavior may be a recommended sleep-improving intervention for those with sleep problems. We included the following text in the Limitations:

„During recruitment to BSETS, we prioritized sample size and high adherence to demographic representativity. As a result, BSETS participants are mainly healthy young participants, and its main goal is to study the effects of common daily experiences on normal sleep. BSETS findings may not automatically generalize to an elderly population or to those with sleep disorders. However, this limitation can be overcome in BSETS-based studies by using between-individual variation as a moderator. In other words, we can investigate if a certain relationship between DAE and sleep is different as a function of demographic (e.g. age and sex) or health-related (e.g. trait depression, overall sleep quality) characteristics. Furthermore, BSETS data can be used to identify DAE with effects on normal sleep, which can then be used in targeted interventions in more focused studies of the elderly or those with sleep disorders.”

(2) The presented questionnaires (Sect. 2.3.2) are well designed. However, did you investigate the amount of caffeine intake, the presence of medication of any kind (targeting cardiovascular or breathing disorders, neurological disorders, psychological disorders) or the presence of familiarity with any neurological disorder? Cardiovascular impairment or neurological disorders may significantly impact the sleep pattern, resulting in higher presence of spindles, arousals or high sleep fragmentation. Please discuss.

Reply: Unfortunately we did not include questions about caffeine use in our questionnaire. We did so because of practical length limitations to our questionnaires limited how many daily experiences we could ask about, and because the effects of caffeine on sleep are well known (see e.g. Clark & Landolt 2017 and Gardiner et al 2023) this presented a less likely area where BSETS could produce novel results. We included this in our limitations.

We did, however, collect data about medication use. As we state, „We collect data about daily medication use, permitting the post-hoc exclusion of participants seriously affected enough to be medicated”.

(3) In Section 2.3.3. the PSD is introduced as elected quantitative metric for the evaluation of eeg. Later on, in Section 4.3, you mention that the lack of central channels may lead to an underestimation of the presence/number of sleep spindles (and sigma rhythm). In view of this, does the PSD still remain a reliable metric, or there may be the need for including other quantitative assessments retrieved from the EEG? Indeed, morphological (i.e., time-domain) characteristics are also widely investigated in literature. I believe this is a key point to be discussed.

We acknowledge that the lack of topographic representation of channels in our mobile EEG system is a limitation to our analyses. In order to estimate the severity of this limitation, we used data from 203 adult (age>18) participants with standard full-night polysomnography to estimate the similarity of PSD estimates on frontal and other channels. Details are provided in a new S1 Text (excerpt):

“Briefly, we used scalp EEG data from a large sample of 203 adults (Ujma et al. 2023) with all-night laboratory polysomnography data with 10 channels. (See the original reference for additional recording details.) We calculated correlations between PSD estimates on two frontal (Fp1, Fp2) and eight other (C3, C4, F3, F4, O1, O2, P3, P4) channels to estimate how well non-frontal PSD estimates are approximated by those available in BSETS. The findings are graphically depicted below.

Between-participant Pearson correlations between PSD estimates obtained from Fp1/F2 and eight other EEG channels.

We found reasonably high correlations, usually in the order of 0.7-0.8. Local minima are observed at low frequencies in REM (rmin∼0.4), and in the fast spindle frequency range in NREM (rmin∼0.55), in line with the topographic specificity of some EEG waveforms observed at these frequencies (most likely, eye movements and fast spindles). 

These findings confirm that frontal PSD estimates are an imperfect, but reasonably accurate proxies of PSD estimates on other channels. Frontal channels in BSETS likely have considerable utility in detecting EEG events with non-frontal maxima.

We also included the following additional text in the main body of the manuscript:

“Quantitative EEG analyses in BSETS are limited by the frontal topography of available EEG channels. We quantitively assessed the scope of this limitation by comparing PSD estimates from frontal and other channels in an external dataset with a full PSG cap (Text S1). We found that between-participant correlations of PSD estimates between frontal and other channels are reasonably high (typically r=0.7-0.8). A local minimum is observed in the fast spindle frequency range, but even here a minimal correlation of >0.5 is observed. This suggests that the frontal PSD estimates in BSETS are reasonable proxies for activities which occur without a characteristic frontal topography. “ 

We also included a brief summary of these findings in the Limitations section of our paper.

We also agree that morphological characteristics (such as individually detected sleep spindles and slow waves) are interesting features to be extracted from BSETS EEG data. However, existing algorithms may not be applicable for mobile EEG systems such as Dreem2, and the development of new detection methods was beyond the scope of the current paper. We now mention the detection of morphological features and their association with traits and daily experiences as possible future research opportunities.

---

## [Decision Letter · Decision Letter 1]

7 Jul 2023

Protocol of the Budapest Sleep, Experiences, and Traits Study: an accessible resource for understanding associations between daily experiences, individual differences, and objectively measured sleep

PONE-D-23-10162R1

Dear Dr. Péter,

We’re pleased to inform you that your manuscript has been judged scientifically suitable for publication and will be formally accepted for publication once it meets all outstanding technical requirements.

Kind regards,

Luigi Borzì, Ph.D.

Academic Editor

PLOS ONE

Additional Editor Comments (optional):

The authors have carefully addressed all the comments and suggestions from the Reviewers and the Editor.

The manuscript can now be accepted for publication.

Reviewers' comments:

Reviewer's Responses to Questions

**Comments to the Author**

1. Does the manuscript provide a valid rationale for the proposed study, with clearly identified and justified research questions?

Reviewer #1: Yes

Reviewer #2: Partly

2. Is the protocol technically sound and planned in a manner that will lead to a meaningful outcome and allow testing the stated hypotheses?

Reviewer #1: Yes

Reviewer #2: Partly

3. Is the methodology feasible and described in sufficient detail to allow the work to be replicable?

Reviewer #1: Yes

Reviewer #2: Yes

4. Have the authors described where all data underlying the findings will be made available when the study is complete?

Reviewer #1: Yes

Reviewer #2: Yes

5. Is the manuscript presented in an intelligible fashion and written in standard English?

Reviewer #1: Yes

Reviewer #2: Yes

6. Review Comments to the Author

You may also provide optional suggestions and comments to authors that they might find helpful in planning their study.

Reviewer #1: The authors properly addressed the reviewer's concerns. Accordingly, I do not have additional comments.

Reviewer #2: The Authors have carefully revised the article according to the Reviewers' response, and the structure and technical details are improved.

7. PLOS authors have the option to publish the peer review history of their article (what does this mean?). If published, this will include your full peer review and any attached files.

Reviewer #1: No

Reviewer #2: No

---

## [Editor Report · Acceptance letter]

11 Jul 2023

PONE-D-23-10162R1 

Protocol of the Budapest Sleep, Experiences, and Traits Study: an accessible resource for understanding associations between daily experiences, individual differences, and objectively measured sleep 

Dear Dr. Ujma:

I'm pleased to inform you that your manuscript has been deemed suitable for publication in PLOS ONE. Congratulations! Your manuscript is now with our production department. 

Kind regards, 

on behalf of

Dr. Luigi Borzì 

Academic Editor

PLOS ONE